# Could Failures in Peer to Peer Accommodation Be a Threat to Public Health and Safety? An Analysis of Users Experiences after the COVID-19 Pandemic

**DOI:** 10.3390/ijerph20032158

**Published:** 2023-01-25

**Authors:** Maria del Mar Alonso-Almeida

**Affiliations:** Department of Business Organization, Faculty of Economics and Business Administration, Universidad Autónoma de Madrid, 29049 Madrid, Spain; mar.alonso@uam.es

**Keywords:** service failures, Airbnb, P2P recovery, P2P accommodation failures, strategy, P2P service quality

## Abstract

Peer to Peer (P2P) accommodation has effected a disruptive change in cities and traditional tourist destinations, with considerable growth in the number of customers and hosts offering services worldwide. This growth is not without the problems that arise from service quality failures. Previous research has largely concentrated on positive consumer responses to P2P accommodation, while failures in service have been neglected. Research regarding the impacts of failures on health and safety issues is particularly scarce, especially after the pandemic. Therefore, this research is exploratory in nature and drew on the real experiences of 91 guests worldwide since the beginning the pandemic until 2022. It analyses failures in public health and safety regarding P2P accommodation, classifying them and expanding the categories in order to design recovery strategies to mitigate the negative impacts. The findings provide novel insights that help understand failures in P2P accommodation from a guest’s perspective. Moreover, this research suggests recommendations to platforms and hosts that will help improve the levels of service quality and trustfulness for this type of accommodation.

## 1. Introduction

Airbnb is considered a successful business; this is demonstrated by the number of results yielded when one types “Airbnb” into Google: more than 228,000,000. A search performed using Google Scholar yields 41,200 documents when ‘Airbnb’ is entered, and a search on “P2P short rentals” yields more than 23,000 results. Thus, a great amount of literature regarding Airbnb has been developed, mainly by guests and hosts, regarding the availability of Airbnb units and their impacts on destinations, regulations, their impacts on the tourism sector, and on Airbnb the company [1].

Figures also show the success this innovative business model has achieved in only 15 years. In fact, there are currently more than 4 million Airbnb hosts worldwide and more than 6 million listings on the platform, with more than 150 million worldwide users who have booked more than 1 billion stays and more than 100,000 cities worldwide that have active Airbnb listings [2]. 

This worldwide growth has resulted in disruptive changes in cities and traditional tourist destinations, provoking mixed sentiments from the stakeholders involved [3,4]. In regard to destinations, among the positive impacts are increases in the number of visitors to a particular region; the creation of new entrepreneurs and employment opportunities; a decrease in the informal economy; improvements in public infrastructures and transportation; and an enabling of a circular economy [5,6]. On the other hand, the most frequently reported negative impacts are the displacement of long-term rentals by short-term rentals and an increase of gentrification [5]; an increase of noise, waste, and unfamiliar people in residential areas; communal area damages; and an increase in insecurity and disturbances of privacy [6].

For hosts, P2P accommodations provide an opportunity to obtain extra income, create their own job, use assets that otherwise would have been wasted, and improve their standard of living [7]. For guests, the accommodations provide an opportunity to experience an area as residents do; gain access to more flexible amenities than in a hotel; and promote sustainable consumption. Such accommodations also are a good financial choice compared with the alternatives [8,9]. 

Previous research has largely concentrated on positive consumer responses taken from online reviews on the Airbnb platform or Airbnb user surveys in analysing P2P accommodation service quality, whereas problems with failures in the service have been neglected [10]. Research regarding the impacts on guests of failures in health and safety issues and their consequences have been particularly neglected. According to our best knowledge, the scarce research on this topic—public health and safety—has focused on analysing the safety amenities available at P2P accommodations [11]; the health and safety experiences of Airbnb hosts and guests, with a particular focus on risk assessment and risk management [12]; or on perceptions of the differences in safety and security between traditional licensed properties and Airbnb accommodations [13]. 

More in-depth research is needed to determine the typology and impact of health and safety failures on P2P accommodations, given that this issue is underresearched, especially after the COVID-19 pandemic. Thus, this research had a twofold purpose: (1) to identify the main health and safety failures in P2P accommodations after the pandemic and (2) to classify them into categories that extend the previous classifications done by [8] on the basis of Airbnb histories from guests. These goals reveal new insights on health and safety failures in P2P accommodations and provide potential strategies to face them in times of crisis.

This article is organised as follows. In the next section, we present a review of the literature. This is followed by a description of the method that was used for the empirical study and a qualitative analysis of the data. We conclude the article with an analysis of the results, a presentation of the conclusions, and suggestions for future research.

## 2. Literature Review

### 2.1. Assessment of Health and Safety on Peer to Peer Accommodation before the COVID-19 Pandemic

As [14] asserted, Airbnb has created a leading P2P accommodation network based on the digital technology in two-way markets linked to hosts and travelers. The company has endorsed the philosophy and strategies of a circular economy by means of sharing underused places—rooms and/or houses, in a wide sense—thus transforming the tourism sector and tourists’ images of various destinations.

Service quality is a very important dimension of services. [15] created a quality service model called SERVQUAL that is often used in a number of sectors, including tourism. This model includes five dimensions of quality: (1) reliability, (2) responsiveness, (3) security, (4) empathy, and (5) tangible elements. In P2P accommodations, both the platform and the host are involved in the quality of service, forming a triad of services and interactions whereby the measurement of service quality is more complex and involves more factors.

The hour of truth comes in the delivery of the service: the accommodations. The pre-trip information provided by the host creates an image of the accommodations in the mind of the traveler. Differences between that image and the real accommodations could generate a perception of low service quality and dissatisfaction. 

Although the most of the research has focused on the positive impact of P2P accommodations, a small number of researchers have studied the motivations of dissatisfaction. Ref. [16] examined dissatisfaction with both the platform and the accommodations and noted that the main sources were information quality and service quality. Travelers reported a lack of congruence between what is published on the Airbnb site and the reality of the accommodations. As mentioned, most dissatisfaction stemmed from a perception of failure in service quality [15,16]. In fact, [17] found a large gap between guests’ expectations of P2P accommodations and the reality of those accommodations.

The worst failures are produced when the delivery of service fails. Depending of the severity of these failures, guests can decide to discontinue that service [18]. In line with this assertion, [10] analysed the drivers of Airbnb discontinuances and found two categories of motivations: (1) online service issues related to the platform and (2) offline service related to the accommodations and the host’s behaviour. The biggest drivers of discontinuance in both online and offline service are related to accommodations and hosts before and after accommodation delivery. In fact, guests’ perception is that accommodation is not standardised. Thus, the inability to control host service quality is one of the most reported complains of Airbnb guests. 

Failures in sanitation and safety issues have varying levels of severity and inconvenience for the guests. Before the pandemic, [12] had already reported health risks related to Airbnb P2P accommodations; problems of cleanliness, such as dirt in the bathroom and kitchen, used sheets on beds, leftover food in the fridge, or dust and cobwebs. Cleanliness is a crucial determinant of guest satisfaction and a lack of cleanliness is considered a severe failure [19]. In addition, [12] warned that a P2P accommodation with sanitation problems could be a health risk for guests (e.g., a source of infection). Therefore, P2P accommodations with sanitation problems could pose serious health hazards. 

In the same line, [10] pointed out a large range of sanitation issues that can affect one’s health, such as a lack of cleanliness or hot water, as well as broken facilities (e.g., fridge, air conditioner, heating, toilets, bed, shower). Moreover, those researchers identified that some accommodations presented problems such as “accommodation was ‘filthy’ ‘mouldy’ and ‘smelly’, and sometimes infested with cockroaches, fleas, ants, spiders and bedbugs” [10] (p. 102798). This type of P2P accommodation could present serious health hazards. 

Sanitation issues are not the only problems reported. In fact, “safety is a major concern when choosing Airbnb accommodation” [10]. In particular, personal dangerous internal housing situations, such as gas leaks and broken glass, or external ones, such as neighborhood location, drugs lying around, crime, violence, or places of construction were cited by guests as the main safety threats [12]. 

Another category of threats to personal security is the perceived risks provoked by hosts [20]. For example, when a host enters the accommodation without permission while the guest is inside [8]. These situations pose a threat to personal safety and security. Situations such as when a host enters the guest room or house, or presents an awkward situation for the guest have been reported, especially by women [10]. In fact, some female guests asserted that have experienced sexual harassment or assault while they stayed at Airbnb accommodations while traveling alone [12]. Conflicting situations among hosts and guests are also a threat; some hosts have become violent [21]. 

These situations are considered severe failures in service and elicit high levels of negative emotions, even in people with experience using P2P accommodations [21]. These situations could modify travellers’ intention to use and recommend P2P accommodations [22], with negative consequences for tourism destinations and cities.

### 2.2. Assessment of Health and Safety on Peer to Peer Accommodation after the COVID-19 Pandemic

Several studies explain that travelers’ behaviors seem to have changed following the lockdowns; reducing the number of long trips, travelling near home and increasing the proportion of travelling with family to avoid social contact [3].

As [3] explained, the perception of risks has an objective dimension and a subjective dimension based on multiple elements of the real context and physical characteristics of the accommodation. Health and safety risks have been exacerbated due the pandemic and enter into a new era in the consumer relevance [23]. The perception of risks in relation to health and safe practices has increased when tourists visit places, facilities, transportation, and accommodations, provoking new demands for hygiene and personal safety [24]. Therefore, the pandemic is likely to improve the health and safety standards of P2P accommodations and take the industry to a higher level. 

Cleanliness information provided by hosts, by the Airbnb platform, and by previous guests’ reviews have significant effects on customers´ trust [23]. Nevertheless, failures of sanitation in accommodations seem to be a common complaint for P2P accommodations despite Airbnb’s claims that it has implemented improved cleanliness requirements and safety and security protocols [25], and stated that cleanliness had become a top priority after the pandemic lockdowns [26]. This cleanliness protocol was transformed by mandate for all their listing properties worldwide, with the exception of mainland China [27]. The biggest complaint when any sanitation or safety issues arise is Airbnb’s limited ability to resolve these situations, though it has a safety team devoted to pursuing suspicious activities.

The importance of cleanliness and safety in P2P accommodation has become so relevant after the lockdowns that “the pandemic has created more awareness of a clean environment, which, in turn, altered guest attitudes and behaviors” [23]. In fact, [28] pointed out that the pandemic does not uniformly affect all guests. Its impact depends of the pandemic anxiety of the guests. Therefore, guests could be more sensitive to failures in sanitary and safety measures than before the pandemic and require more information regarding the protocols of sanitation and cleanliness that had been followed before and after the stay.

Ref. [29] classified tourist perceived safety according to four dimensions: human elements; facility and equipment elements; environmental elements; and management elements. In the case of safety, social distance aims to keep people from becoming infected. Thus, people are more cautious and pay more attention to maintaining social distance from others [30]. This suggests that P2P accommodations could be perceived as dangerous and that people would not be comfortable sharing accommodations. This means that the previous P2P accommodation strengths could be considered weaknesses after the pandemic [24]. One first change is the increase in the frequency of entire house host selection instead of just renting a room, and the desire for less host and community interaction [23]. 

The smooth operation of amenities is more relevant than before the pandemic due to the fear of contact with strangers [31]. The same situation is reported for the location as a whole. Guests avoid city centers and seek less crowded tourism destinations and places with low levels of COVID-19 contagion and high levels of vaccination [32]. Finally, management risks are related with protocols and additional measures to promote trust, such as cleanliness protocols or remote procedures to resolve problems without personal contact from hosts, among others [33].

## 3. Methods

### 3.1. Sample 

Information was gathered from AirbnbHell (https://www.airbnbhell.com/, accessed on 9 December 2022) following the method called passive lurker format [34], which has been used by a growing number of researchers (e.g., Ref. [10] or [18]).

The passive lurker method involves the use of content that exists in the public domain that has been generated by users (i.e., the content is user generated). This method has several benefits for investigation [18]. First, it is not intrusive at all. Second, it is provided by the user spontaneously, without any requirements, thus minimising researcher and participant bias. Third, it is cheap and very convenient because it grants the researcher access to user-generated content worldwide that normally is very expensive and very difficult to obtain. Therefore, it saves time, allowing for exploration of the phenomenon in a timely manner. Finally, no user consent is needed for the data because they are available in the public domain. 

AirbnbHell contains real, uncensored Airbnb stories from hosts, guests, and neighbours. The site is 100% independently owned and operated and is not owned by, or affiliated with any competitors of Airbnb.com [35]. Histories are a better resource than just Twitter comments used by authors like [10], because they contain more data to explain the situation than just the 140 characters permitted on Twitter. This type of content allows data that are more descriptive, rich, and detailed. In addition, AirbnbHell allows for the inclusion of photos, images of WhatsApp messages or Airbnb screens, and users can provide links to the accommodations involved. This is relevant for collecting a deep and complementary written history because visual stimuli such as photos help clarify the ambiguity of a written scenario [36]. 

The histories we selected for this study were written between the beginning of the COVID-19 pandemic in March 2020 until October 2022, in order to cover the entire COVID-19 period. A total of 91 online histories were gathered; in total, 47,492 words and 208,533 characters without spaces. Each history had an average of 522 words. In addition, the histories contained 179 photos; 30 images of messages between the guests and Airbnb or hosts, 77 images, and 31 links from the exact P2P accommodation where the service failure happened.

Seventy-seven of the guest histories were from North American P2P accommodations, the 13% were from Europe, 7% were from Latin America, and 3% were from other regions. Less than 1% of guests asserted that the failure they were reporting involved their first P2P accommodation experience; in fact, the most of guests reported previous multiple positive P2P accommodation experiences before the pandemic. Finally, 10% noted that they also were a host or superhost. 

### 3.2. Methodology

An inductive and deductive qualitative content analysis approach to analyse users’ histories to gain a more complete understanding of the topic was used. 

The following steps were taken. First, we analysed the guests’ histories using a thematic analysis based on the categories found by [36] with regard to the P2P accommodation and host—offline service issues—and the perceptions of a lack of safety and security found by [6] regarding P2P accommodations. Second, we used lexicon techniques, such as semantic logic, word combinations, and frequency of words with interconnected categories in different circumstances. This method has been used in content analysis to effectively filter information [37]. Third, adhesion to informed categories without restrictions were run to identify emerging topics [38]. Finally, we combined the categories with the topics identified so we could validate, refine, and expand the thematic categories. 

## 4. Results and Discussion

Each history included one or more categories of health and safety issues, but three emerged most frequently: (1) Undesirable host attitudes and/or behaviours (100%), (2) Sanitation issues and/or amenity malfunctions (87%), and (3) Safety and security concerns (30%). The other categories—No service delivery guarantees, Low comparative performance, and Accommodation of low value for the money—even though they are important when evaluating P2P accommodation service, are not relevant for the goals of this research. 

Multiple category chains contained the following relationships, in this order: (1) Undesirable host attitudes and/or behaviour–Sanitation issues and/or amenity malfunctions, (2) Undesirable host attitudes and/or behaviours–Safety and security concerns, (3) Sanitation issues and/or amenity malfunctions–Safety and security concerns, and (4) Undesirable host attitudes and/or behaviours–Sanitation issues and/or amenity malfunctions–Safety and security concerns.

We split some of the categories to focus on each aspect of the category in depth. Thus, Sanitation issues and/or amenity malfunctions and Safety and security were separated into two categories. Sanitation issues and Amenity malfunctions were split into two categories. Safety and security concerns was also divided into two categories: (1) Safety and security inside and (2) Outside of P2P accommodations. 

Upon examining the information more deeply, new top-down categories emerged. The results of health threats found in the Sanitation issues category are depicted in Figure 1. The strongest dimension related with P2P accommodation failure was dirt. Thus, dirt was associated with two subcategories: (1) Dirty place and (2) Type of dirt. 

Complaints about unsanitary conditions in an accommodation mentioned dirty places—kitchen, toilet, dining room, and bedrooms—with more intensity in kitchens and bathrooms. In the case of type of dirt, five types emerged, with varying levels of severity. The circles in Figure 1 are sized according to the frequency of the type of dirt mentioned in the histories.

The biggest complaint was related to wetness, mold, and black stains, all of which are linked to failures in infrastructure and bad maintenance. This is a severe problem that can be dangerous for people’s health, especially those with respiratory diseases. Some guests’ comments regarding this subcategory are:
“*When I used the washer, the hose did not drain and it was loose and flooded the entire apartment. Every time it rained there were multiple leaks around the unit.*”
“*There was (…) mold on the carpet.*”
“*We immediately noticed black mold covering the entire HVAC system and surrounding closet within the interior of the property.*”

The second type concerned dirt that could be associated with the normal use of an amenity. Dust or spiderwebs is a signal of poor cleaning care that seems to be a normal failure in P2P accommodations. Some previous authors and guests have associated this failure with a lack of cleanliness standards, even after Airbnb had established a specific cleanliness protocol [17]. Regarding this subcategory, some experiences are:
“*The place was very dirty, having paw marks in the dust under the bed along with random socks and other trash, and even a black hand mark on the mattress.*”
“*All over the apartment were spiderwebs.*”
“*There were old bars of soap in the tub area, splatters in the toilet as if someone just had diarrhea, toothpaste on the mirror and hair on the floor.*”
“*The bathroom was disgusting with a moldy shower curtain and orange slick grime.*”

The third type of dirt—bad smell and food leftovers—is similar to the previous one, but related to the kitchen and bathroom:
“*It smelled like a medieval pub.*”
“*The countertops had old food debris and needed to be wiped.*”
“*Exploded food left by others in the microwave.*”

These two types of dirt are associated with very negative emotions, such as “our disgusting rental unit” or “this has been an incredibly upsetting experience for us”.

The fourth type of dirt brought together body fluid stains, especially in sheets, bed covers, mattresses and sofas. It is a severe failure and by itself supposes a very unpleasant situation for guests, independent of the other failures:
“*It was a dump: filthy walls, cobwebs, cupboards falling off the hinges, rusted out washer and dryer, broken dishwasher not attached to the cupboard, all three showers broken, filthy stained couches, soiled mattress with what looked like urine and blood stains, patio furniture covered with blankets to hide the stains.*”
“*There was blood on the fireplace glass, mold on the carpet, open containers in the refrigerator, urine on the bed hidden with a sheet, who knows what in the shower, and these were just a few things.*”
“*The snow began to melt and when it did there was about 20 piles of dog feces all over the deck, becoming unsanitary.*”

Finally, the fifth type of dirt was associated with pests:
“*The place was infested with cockroaches.*”
“*There were multiple live cockroaches.*”
“*It was the final straw when I went to brush my teeth at 7:00 a.m. and a cockroach and four baby cockroaches came out of the drain in the sink.*”
“*I contacted Airbnb about this bedbug problem on March 2. It is currently March 11 and I am sitting in a hotel, still with no answer about this problem.*”
“*Being bit[ten] by fleas or bedbugs as soon as we lay on top of the bed.*”
“*My sister got a zip lock bag and put the live bedbug in it.*”

This is also a severe sanitation problem and it may pose a public health problem, as some guests asserted:
“*It was disgusting and a health hazard.*”
“*It was definitely not COVID compliant.*”

In addition, these conditions provoked very negative emotion descriptors, such as “terrible”, “nightmare”, or “horrible experience”.

Regarding the COVID-19 disinfection protocols promised by Airbnb, guests reported that they were not followed in these accommodations.

Several authors asserted that the location of the accommodation, the amenities state and building itself are a relevant part of the guest satisfaction [39]. Thus, any failure could consider a dissatisfaction situation and depending of the failure a danger for guest´s safety and security. With regard to amenity malfunction, three subcategories emerged: (1) Lack of maintenance on the amenities, (2) Broken, and (3) Not working. The more stressed resulted being the first one (see Figure 2). 

In cases of lack of maintenance, guests reported serious deterioration in home infrastructure in general, with aged amenities and broken, unfixed items. This situation can produce accidents such as cuts or falls, among other things:
“*The garden area was untidy, with the grass not cut and loose tiles on the path.*”
“*There are broken or dirty items (linens with blood stains and a foam mattress.*”
“*The dishwasher was rusty with broken parts; the rust stayed on the presumably washed dishes afterward.*”
“*The place was hot, like 100 degrees F hot (+30 Celsius), and all three bedrooms did not have AC. There were two ancient AC units in the living room and dining room.*”
“*We could see the door lock on the security gate wasn’t installed and there was a gap.*”
“*The floor was chipped and cracked and the window screen was broken.*”

Moreover, broken amenities constituted a severe cause of dissatisfaction for guests that could contribute to health problems or accidents. For example, a broken leg on a bed could cause insomnia, or a broken chair could cause someone to fall. Thus, some of the guests’ comments were:
“*The stained mattress was still in place, and there was a broken dishwasher and cupboards.*”
“*Broken or dirty items (linens with blood stains and a foam mattress).*”
“*We had a broken toilet seat.*”
“*I checked the bed, and noticed the middle legs were broken.*”
“*A stack of bricks [were put] on one corner of one of the beds to replace a broken leg.*”

Finally, nonworking amenities could cause frustration for guests because one of the main motivations for choosing a P2P accommodation is the inclusion of amenities not available at a hotel [12]. Some examples of nonworking amenities were:
“*Nonworking TV.*”
“*The washer and dryer didn’t work.*”
“*The washer was not working.*”
“*The air conditioning on the first floor was not working.*”

Different types of nonworking amenities had different levels of severity and impact. Broken items such as a cooker or a TV can be a motivation for anger because guest expectations are not met, while other nonworking items can pose a risk of infection and become a health threat. 

As mentioned, safety and security concerns were also divided in two categories: (1) Safety and security inside and (2) outside of the P2P accommodation. Figure 3 shows the results obtained. 

The safety and security inside the P2P accommodation seems to be related to some threats from outside. The first emergent category was the host. Hosts can be a hazard for the guest when their behaviour or attitude is not appropriate. In fact, guests reported hosts’ aggressive and other negative attitudes that can lead to a nervous and stressful situation inside and outside the accommodation:
“*As we were listening to him, another man who did not identify himself and was dressed in track pants and t-shirt came out of nowhere in a very aggressive way and started demanding that I, a female, leave our room and go with him to the front gate of the property to show him the code we had tried to use that wasn’t working.*”
“*The maintenance worker put his hand on this man’s shoulder to hold him back and calm him down because he was acting so aggressive and uncontrollable.*”
“*Host was over aggressive.*”

The most frequently pointed out actions regarding safety were violations of privacy, such as registering the guest’s baggage, entering the guest’s room or house when the guest is outside, and recording guests in their rooms:
“*I find it disappointing that Airbnb would be okay for a host to have an active recording camera in the living room, violating one’s privacy, and the fact that they find it okay with a host having a listing that nasty.*”
“*Unauthorised people are just letting themselves onto the property, opening the backyard gate and walking in, totally ignoring me, but, once confronted, they run away. I’ve rented this property for a month and what privacy do I get? These trespassers are jeopardising the well-being of my beloved pooch from escaping. They are trespassing on private property and leaving the backyard fence open while doing so.*”
“*I always put my zipper on my backpack or suitcase in a certain position. It had moved. He didn’t take anything, because I took all my valuables with me in a second backpack. But that’s a huge violation. The only lock that worked on the bedroom door was a keypad lock that he said didn’t work, but I didn’t know if he could put in a battery from the outside and try to get in. When I came in the doorknob was loose and I couldn’t turn it to get back into my room. I kept turning until it tightened and I was able to get in. Because of the lock, I had to put a table against the door and sleep in my clothes, all packed in case I had to leave in a moment’s notice.*”

Regarding inside P2P accommodations, another three subdimensions arise that pose health hazards, such as allergic reactions provoked by dust or insects bites. A second type of health hazard is injuries by cuts from broken mirrors. As mentioned, these threats are linked to sanitation issues. The third subcategory was illicit acts regarding P2P accommodation, including more guests than allowed, subleasing a property, or asking a guest to lie.
‘*He lied. There were multiple codes to enter the elevator and the unit that didn’t work all the time either. There were also signs posted at the elevators stating that they do not allow Airbnb.*’
‘*The Airbnb Plus property is located in a building that does not allow short-term rentals. We had to constantly lie to the concierge and every time we parked our car in the garage.*’
‘*I am at a loss how people can be like this and rip people off and lie so blatantly.*’

It is possible to assert that inside the P2P accommodation, guests could have safety and security risks of different types, as previous research has suggested.

There are also safety and security hazards outside P2P accommodations; specifically, two dimensions emerged: (1) safety and security hazards in the neighborhood and (2) animals. In relation to the neighborhood, multiple hazards were reported, such as drugs, robbers, or violence, which caused fear and insecurity as well as insalubrity in the surroundings. This analysis shows the other face that [7] revealed. These authors studied the neighbors’ perceptions of P2P accommodations and provided information on locations of some P2P accommodations. Thus, guests complained that:
“*We did not want to endanger their lives by going to our Airbnb with active threats of violence in the area we would have to cross through.*”
“*The neighborhood is trashy, unkept, and very noisy with neighbors consistently slamming doors, babies crying, chickens crowing at all hours of the day, as well as other things.*”
“*When we arrived we were shocked and disturbed. The apartment was in an unsafe-looking neighborhood in what looked like Section 8 housing.*”
“*We noticed a homeless person screaming and doing her business in public.*”
“*[The host] just left out that the home is surrounded by homeless people, tents, trash, and the walk might be one of a lifetime.*”

In regard to animals, guests especially warned about exposure to a dangerous animal, such as dogs or rats:
“*We found [our dog] three days later torn apart by a coyote.*”
“*A crazy drunk neighbor with a dog who bit my child and who killed a hen in front of our eyes.*”
“*Problems with the host and his large dog.*”
“*They had dogs that barked and might bite.*”
“*While spending time at the pool the first day at the property, a huge rodent (rat) ran right by me.*”

Thus, after the full analysis, it is possible to assert that some P2P accommodations have considerable room for improvement in public health and safety issues after the pandemic. Airbnb should gather the complaints, investigate the acts reported, and try to resolve the problems. Previous research [8] has pointed out that Airbnb was identified as having a poor service recovery, with inadequate customer support when problems arise. Corrective actions could improve the perception of Airbnb’s customer service, P2P accommodations, and tourism destinations.

To summarise, dirt, a lack of maintenance, and broken and nonworking amenities in P2P accommodations have been identified as major threats to public health. Severe and continuous failures were noted in different P2P accommodations by different guests and countries. Those findings warn about the necessity to establish some standards for this type of accommodation. In addition, recovery actions and procedures for the most recurrent failures should be defined and developed by Airbnb and adopted by hosts to resolve some problems immediately. 

These findings confirm and reinforce previous research [10,12,13,14,22] regarding failures in public health and safety in P2P accommodation. Moreover, they provide novel insights extending and detailing the typology of failures in the categories of sanitation, amenities, and safety and security both inside and outside the P2P accommodation. Thus, sanitation is more important than before the pandemic, and failure regarding this issue constitutes a big threat and major motivation for discontinuance [10]. 

In the case of safety, inside hazards due to the host and problems at the accommodations are not acceptable. Hosts with repeatedly proven violations and who have been accused of undesirable or abusive behaviours should be expelled from the platform as a signal of compromise among all stakeholders—hosts, guests, local communities, and cities, among others. Moreover, hosts should be mindful of the necessity to show professional behaviour to both guests and platforms. Outside safety issues at P2P accommodations can be more difficult to correct, but they can be mitigated. Airbnb should monitor the location and surroundings of P2P accommodations and, if necessary, put a warning on the listing. 

In conclusion, these findings can help foster an understanding of the new necessities of guests after the pandemic and can help establish recovery strategies to avoid becoming a threat to public health and safety and to continue being a successful business model that is profitable for all stakeholders involved, including the tourism destinations themselves.

## 5. Conclusions

Based on the rsults, this study reaches several important conclusions that are relevant to business management, tourism destinations, urban planners, and academics. All of the negative findings about public heath and safety imply very negative emotions on the part of guests, such as being stressed, upset, angry, and having nerves or irritation [40], as well as other negative emotions, such as sadness, disappointment and a feeling of having been deceived. These emotions arise because travel is one of the most frequently mentioned activities as people attempt, after two years of pandemic-related lockdowns, to return to normality in terms of mobility and vacationing for pleasure and enjoyment, with a consequent positive impact on physical and mental well-being [41]. 

This research extends the previous dimensions of research on public health and safety threats in P2P accommodations after the pandemic and provides novel insights to help us understand failures in those services from a guest’s perspective, although corrections of these issues have not been sufficiently studied yet. Such factors should be included in successive research to deepen understanding of the topic.

To protect public health and safety, more control of P2P accommodations should be exerted both by the local governments where they are located and by their own platforms. We propose some solutions.

First, a more trustful host rating system on the platform seems to be an urgent necessity. Many guests report severe discrepancies between the of accommodation information published on a platform and the actual situation. A frequent complaint is that “the host is providing misleading information on the post”. Another complaint is that guests are provided the actual location of the accommodations only after the reservation has been made. Both situations could put guests into trouble regarding health and safety threats by withholding realistic, testable information.

Second, some hosts had very undesirable attitudes and/or behaviours. This is intolerable, not only for Airbnb, but also for cities. Such hosts give a bad reputation not only to the platform but also to the city and its touristic position. This situation reinforces the idea that soft regulation and some minimally proven standards should be imposed on each accommodation when it joins the platform. Moreover, P2P accommodations should maintain these standards beyond the platform rating system, in particular because the Airbnb rating system could be under question due to several guests’ assertions that Airbnb had deleted their negative opinion and evaluation or never made it visible. As a consequence of this practice, most hosts obtain a high rating on the platform.

Finally, our findings provide relevant information for tourism destinations. A tourist destination’s brand could be damaged by an increase of failures by P2P accommodations. Thus, they cannot look the other way or close their eyes while they continue to offer services that can endanger public health. Specific requirements and controls should be required both for accommodations as well as their associated platform, particularly because a lack of control could be derived from problems, as [5] or [6] reported.

This study has some limitations, such as the generalisability of the results, which are derived from the nature of the methodology used. Therefore, in the near future, more information should be gathered to compare health and safety threats before and after the pandemic, as well as their evolution, including the new services offered by Airbnb as the local concierge partners, to help hosts be more efficient and trustworthy. 

## Figures and Tables

**Figure 1 ijerph-20-02158-f001:**
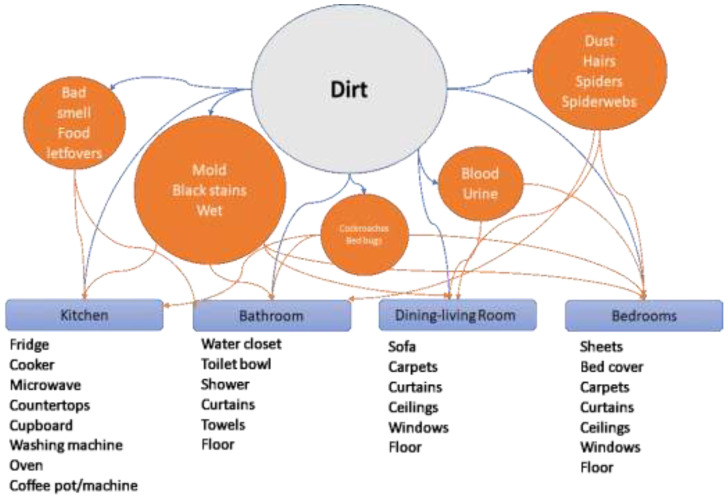
Relationships and categorisation of failures regarding sanitation issues.

**Figure 2 ijerph-20-02158-f002:**
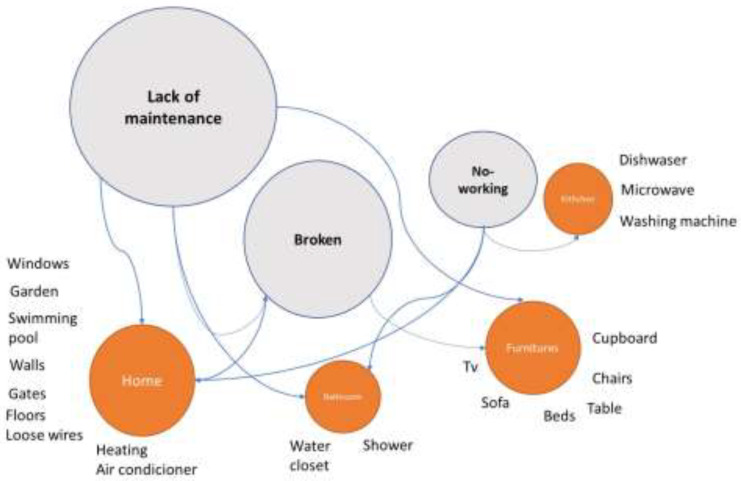
Relationships and categorisation of failures regarding amenities.

**Figure 3 ijerph-20-02158-f003:**
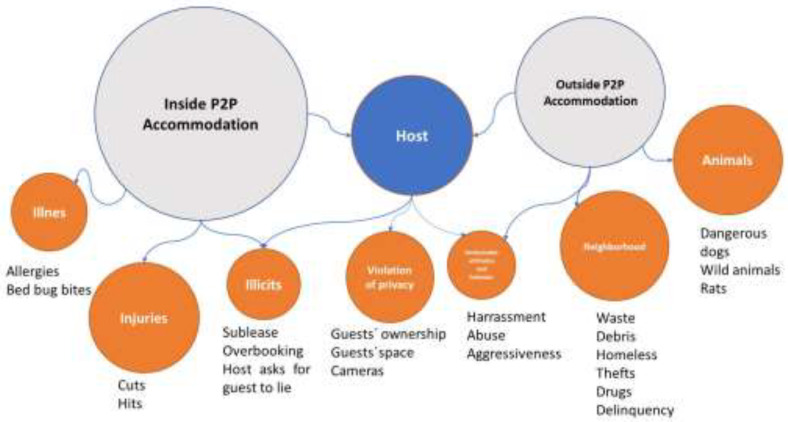
Relationships and categorisation of failures regarding safety and security concerns.

## Data Availability

No applicable.

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
