# Peer review of "Could Failures in Peer to Peer Accommodation Be a Threat to Public Health and Safety? An Analysis of Users Experiences after the COVID-19 Pandemic"

_ijerph, 2023, doi:10.3390/ijerph20032158_

Round 1

Reviewer 1 Report

Dear author,

The article addresses an issue of great relevance to tourism and hospitality, particularly in the growing Peer-to-Peer accommodation market, concerning customer satisfaction and behaviour. However, several issues need to be considered.

Firstly, I suggest using the full meaning of Peer to Peer in the title, not the acronym P2P.

Second, from the manuscript, I doubt the author's literature search capability, because the item of used references is very short.  I suggest the author spend some time on the literature review practice.

Another significant issue is the lack of discussion. The Discussion section should reveal a literature review of studies published or accessed elsewhere. It is worth referring to the international context more extensively. Please, compare your research with similar research so you can be drawn an adequate conclusion. It is crucial in order to raise the scientific level of the manuscript.

Lastly, the manuscript didn’t highlight the influence of Covid-19. So why is the title “An Analysis of Users Experiences during the Covid-19 Pandemic"?

I will be very glad and thankful to see these adjustments.

Author Response

Attached the point-by-point response to the reviewer’s comments.

Reviewer 2 Report

The study is very well carried out and structured. This work is very useful in the field of tourism and accomodation.  I am not the best person to assess the English writing but I was able to read the article in a very comprehensive and understandable way what makes me think that the written is good. However, it should point out some of the territorial implications and research gaps. Perhaps it deserved more bibliographic references and comparison with international studies.

I suggest that the following articles be added in the introduction, namely when it says "This worldwide growth has resulted in disruptive changes in cities and traditional tourist destinations, provoking mixed feelings in the part of the stakeholders involved." (l.35-36). Indeed, the tourism sector has undergone major changes during COVID-19, affecting the very sense of security of individuals visiting cities and using P2P accommodations. I suggest you add the following papers:

https://doi.org/10.3390/su13116399

https://doi.org/10.1108/IJTC-07-2021-0140

Author Response

(The authors gave the same response as above.)

Round 2

Reviewer 1 Report

Dear Authors,

Thank you for the revised version of the paper.

The paper is now more coherent and organised!

The authors have made most amendments based on review comments.

I do not see any warrant objections to its publication.

Reviewer 2 Report

After revisions have been made, the paper is ready for publication.